# Transfer Learning of Graph Neural Networks with Ego-graph Information Maximization

**Qi Zhu**[1],* **Carl Yang**[2],* **Yidan Xu**[3], **Haonan Wang**[1], **Chao Zhang**[4], **Jiawei Han**[1]
[1]University of Illinois Urbana-Champaign, [2]Emory University,
[3]University of Washington, [4]Georgia Institute of Technology
[1]{qiz3,haonan3,hanj}@illinois.edu, [2]j.carlyang@emory.edu,
[3]yx2516@uw.edu, [4]chaozhang@gatech.edu

## Abstract

Graph neural networks (GNNs) have achieved superior performance in various applications, but training dedicated GNNs can be costly for large-scale graphs. Some recent work started to study the pre-training of GNNs. However, none of them provide theoretical insights into the design of their frameworks, or clear requirements and guarantees towards their transferability. In this work, we establish a theoretically grounded and practically useful framework for the transfer learning of GNNs. Firstly, we propose a novel view towards the *essential graph information* and advocate the capturing of it as the goal of transferable GNN training, which motivates the design of EGI (*Ego-Graph Information maximization*) to analytically achieve this goal. Secondly, when node features are structure-relevant, we conduct an *analysis of* EGI *transferability* regarding the difference between the local graph Laplacians of the source and target graphs. We conduct controlled synthetic experiments to directly justify our theoretical conclusions. Comprehensive experiments on two real-world network datasets show consistent results in the analyzed setting of direct-transfering, while those on large-scale knowledge graphs show promising results in the more practical setting of transfering with fine-tuning.[1]

## 1   Introduction

Graph neural networks (GNNs) have been intensively studied recently [29, 26, 39, 68], due to their established performance towards various real-world tasks [15, 69, 53], as well as close connections to spectral graph theory [12, 9, 16]. While most GNN architectures are not very complicated, the training of GNNs can still be costly regarding both memory and computation resources on real-world large-scale graphs [10, 63]. Moreover, it is intriguing to transfer learned structural information across different graphs and even domains in settings like few-shot learning [56, 44, 25]. Therefore, several very recent studies have been conducted on the transferability of GNNs [21, 23, 22, 59, 31, 3, 47]. However, it is unclear in what situations the models will excel or fail especially when the pre-training and fine-tuning tasks are different. To provide rigorous analysis and guarantee on the transferability of GNNs, we focus on the setting of direct-transfering between the source and target graphs, under an analogous setting of "domain adaptation" [7, 59].

In this work, we establish a theoretically grounded framework for the transfer learning of GNNs, and leverage it to design a practically transferable GNN model. Figure 1 gives an overview of our framework. It is based on a novel view of a graph as samples from the joint distribution of its k-hop ego-graph structures and node features, which allows us to define graph information and similarity,

---

*These two authors contribute equally.

[1]Code and processed data are available at `https://github.com/GentleZhu/EGI`.

35th Conference on Neural Information Processing Systems (NeurIPS 2021), Online.

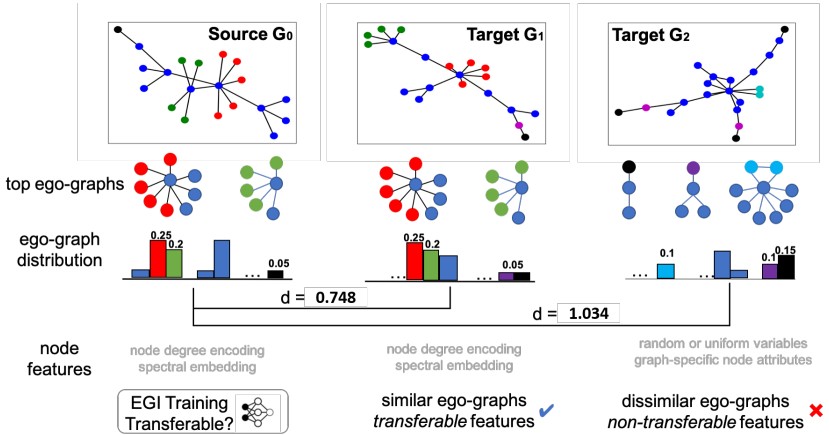

**Figure 1:** Overview of our GNN transfer learning framework: (1) we represent the toy graph as a combination of its 1-hop ego-graph and node feature distributions; (2) we design a transferable GNN regarding the capturing of such essential graph information; (3) we establish a rigorous guarantee of GNN transferability based on the node feature requirement and graph structure difference.

so as to analyze GNN transferability (§3). This view motivates us to design EGI, a novel GNN training objective based on ego-graph information maximization, which is effective in capturing the graph information as we define (§3.1). Then we further specify the requirement on transferable node features and analyze the transferability of EGI that is dependent on the local graph Laplacians of source and target graphs (§3.2).

All of our theoretical conclusions have been directly validated through controlled synthetic experiments (Table 1), where we use structural-equivalent role identification in an direct-transfering setting to analyze the impacts of different model designs, node features and source-target structure similarities on GNN transferability. In §4, we conduct real-world experiments on multiple publicly available network datasets. On the Airport and Gene graphs (§4.1), we closely follow the settings of our synthetic experiments and observe consistent but more detailed results supporting the design of EGI and the utility of our theoretical analysis. On the YAGO graphs (§4.2), we further evaluate EGI on the more generalized and practical setting of transfer learning with task-specific fine-tuning. We find our theoretical insights still indicative in such scenarios, where EGI consistently outperforms state-of-the-art GNN representation and transfer learning frameworks with significant margins.

## 2 Related Work

Representation learning on graphs has been studied for decades, with earlier spectral-based methods [6, 46, 52] theoretically grounded but hardly scaling up to graphs with over a thousand of nodes. With the emergence of neural networks, unsupervised network embedding methods based on the Skip-gram objective [37] have replenished the field [51, 14, 42, 45, 66, 62, 65]. Equipped with efficient structural sampling (random walk, neighborhood, *etc.*) and negative sampling schemes, these methods are easily parallelizable and scalable to graphs with thousands to millions of nodes. However, these models are essentially transductive as they compute fully parameterized embeddings only for nodes seen during training, which are impossible to be transfered to unseen graphs.

More recently, researchers introduce the family of graph neural networks (GNNs) that are capable of inductive learning and generalizing to unseen nodes given meaningful node features [29, 12, 15, 67]. Yet, most existing GNNs require task-specific labels for training in a semi-supervised fashion to achieve satisfactory performance [29, 15, 53, 64], and their usage is limited to single graphs where the downstream task is fixed. To this end, several unsupervised GNNs are presented, such as the auto-encoder-based ones like VGAE [28] and GNFs [35], as well as the deep-infomax-based ones like DGI [54] and InfoGraph [50]. Their potential in the transfer learning of GNN remains unclear when the node features and link structures vary across different graphs.

Although the architectures of popular GNNs such as GCN [29] may not be very complicated compared with heavy vision and language models, training a dedicated GNN for each graph can still

be cumbersome [10, 63]. Moreover, as pre-training neural networks are proven to be successful in other domains [13, 18], the idea is intriguing to transfer well-trained GNNs from relevant source graphs to improve the modeling of target graphs or enable few-shot learning [59, 31, 3] when labeled data are scarce. In light of this, pioneering works have studied both generative [22] and discriminative [21, 23] GNN pre-training schemes. Though Graph Contrastive Coding [43] shares the most similar view towards graph structures as us, it utilizes contrastive learning across all graphs instead of focusing on the transfer learning between any specific pairs. On the other hand, unsupervised domain adaptive GCNs [59] study the domain adaption problem only when the source and target tasks are homogeneous.

Most previous pre-training and self-supervised GNNs lack a rigorous analysis towards their transferability and thus have unpredictable effectiveness. The only existing theoretical work on GNN transferability studies the performance of GNNs across different permutations of a single original graph [33, 34] and the tradeoff between discriminability and transferability of GNNs [47]. We, instead, are the first to rigorously study the more practical setting of transferring GNNs across pairs of different source and target graphs.

## 3 Transferable Graph Neural Networks

In this paper, we design a more transferable training objective for GNN (EGI) based on our novel view of essential graph information (§3.1). We then analyze its transferability as the gap between its abilities to model the source and target graphs, based on their local graph Laplacians (§3.2).

Based on the connection between GNN and spectral graph theory [29], we describe the output of a GNN as a combination of its input node features $X$, fixed graph Laplacian $L$ and learnable graph filters $\Psi$. The goal of training a GNN is then to improve its utility by learning the graph filters that are compatible with the other two components towards specific tasks.

In the graph transfer learning setting where downstream tasks are often unknown during pre-training, we argue that the general utility of a GNN should be optimized and quantified *w.r.t.* its ability of capturing the essential graph information in terms of the joint distribution of its topology structures and node features, which motivates us to design a novel ego-graph information maximization model (EGI) (§3.1). The general transferability of a GNN is then quantified by the gap between its abilities to model the source and target graphs. Under reasonable requirements such as using *structure-respecting* node features as the GNN input, we analyze this gap for EGI based on the structural difference between two graphs *w.r.t.* their local graph Laplacians (§3.2).

### 3.1 Transferable GNN via Ego-graph Information Maximization

In this work, we focus on the *direct-transfering setting* where a GNN is pre-trained on a source graph $G_a$ in an unsupervised fashion and applied on a target graph $G_b$ without fine-tuning.[2] Consider a graph $G = \{V, E\}$, where the set of nodes $V$ are associated with certain features $X$ and the set of edges $E$ form graph structures. Intuitively, the transfer learning will be successful only if both the features and structures of $G_a$ and $G_b$ are similar in some ways, so that the graph filters of a GNN learned on $G_a$ are compatible with the features and structures of $G_b$.

Graph kernels [57, 8, 30, 38] are well-known for their capability of measuring similarity between pair of graphs. Motivated by k-hop subgraph kernels [4], we introduce a novel view of a graph as *samples from the joint distribution of its k-hop ego-graph structures and node features*. Since GNN essentially encodes such k-hop ego graph samples, this view allows us to give concrete definitions towards *structural information* of graphs in the transfer learning setting, which facilitates the measuring of similarity (difference) among graphs. Yet, none of the existing GNN training objectives are capable of recovering such distributional signals of ego graphs. To this end, we design *Ego-Graph Information maximization* (EGI), which alternatively reconstructs the k-hop ego-graph of each center node via mutual information maximization [20].

**Definition 3.1** (K-hop ego-graph). *We call a graph $g_i = \{V(g_i), E(g_i)\}$ a k-hop ego-graph centered at node $v_i$ if it has a k-layer centroid expansion [4] such that the greatest distance between $v_i$ and*

---

[2]In the experiments, we show our model to be generalizable to the more practical settings with task-specific pre-training and fine-tuning, while the study of rigorous bound in such scenarios is left as future work.

*any other nodes in the ego-graph is k,* i.e. $\forall v_j \in V(g_i), |d(v_i, v_j)| \leq k$, *where $d(v_i, v_j)$ is the graph distance between $v_i$ and $v_j$.*

In this paper, we use directed k-hop ego-graph and its direction is decided by whether it is composed of incoming or outgoing edges to the center node, *i.e.*, $g_i$ and $\tilde{g}_i$. The results apply trivially to undirected graphs with $g_i = \tilde{g}_i$.

**Definition 3.2** (Structural information). *Let $\mathcal{G}$ be a topological space of sub-graphs, we view a graph $G$ as samples of k-hop ego-graphs $\{g_i\}_{i=1}^n$ drawn* i.i.d. *from $\mathcal{G}$ with probability $\mu$, i.e., $g_i \overset{\text{i.i.d.}}{\sim} \mu \; \forall i = 1, \cdots, n$. The structural information of $G$ is then defined to be the set of k-hop ego-graphs of $\{g_i\}_{i=1}^n$ and their empirical distribution.*

As shown in Figure 1, three graphs $G_0$, $G_1$ and $G_2$ are characterized by a set of 1-hop ego-graphs and their empirical distributions, which allows us to quantify the structural similarity among graphs as shown in §3.2 (*i.e.*, $G_0$ is more similar to $G_1$ than $G_2$ under such characterization). In practice, the nodes in a graph $G$ are characterized not only by their k-hop ego-graph structures but also their associated node features. Therefore, $G$ should be regarded as samples $\{(g_i, x_i)\}$ drawn from the joint distribution $\mathbb{P}$ on the product space of $\mathcal{G}$ and a node feature space $\mathcal{X}$.

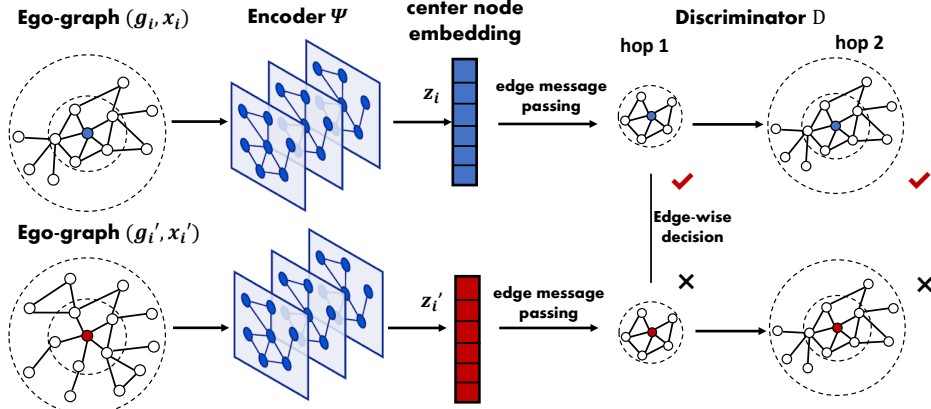

**Figure 2:** The overall EGI training framework.

**Ego-Graph Information Maximization.** Given a set of ego-graphs $\{(g_i, x_i)\}_i$ drawn from an empirical joint distribution $(g_i, x_i) \sim \mathbb{P}$. We aim to train an GNN encoder $\Psi$ to maximize the mutual informaion (MI $(g_i, \Psi(g_i, x_i))$) between the defined structural information $g_i$[3] (*i.e.* k-hop ego-graph) and node embedding $z_i = \Psi(g_i, x_i)$. To maximize the MI, another discriminator $\mathcal{D}(g_i, z_i) : E(g_i) \times z_i \to \mathbb{R}^+$ is introduced to compute the probability of an edge $e$ belongs to the given ego-graph $g_i$. We use the Jensen-Shannon MI estimator [20] in the EGI objective,

$$\mathcal{L}_{\text{EGI}} = -\text{MI}^{\text{(JSD)}}(\mathcal{G}, \Psi) = \frac{1}{N} \sum_{i=1}^N \left[ \text{sp}\left(\mathcal{D}(g_i, z_i')\right) + \text{sp}\left(-\mathcal{D}(g_i, z_i)\right) \right], \tag{1}$$

where $\text{sp}(x) = \log(1 + e^x)$ is the softplus function and $(g_i, z_i')$ is randomly drawn from the product of marginal distributions, *i.e.* $z_i' = \Psi(g_{i'}, x_{i'})$, $(g_{i'}, x_{i'}) \sim \mathbb{P}$, $i' \neq i$. In general, we can also randomly draw negative $g_i'$ in the topological space, while enumerating all possible graphs $g_{i'}$ leads to high computation cost.

In Eq. 1, the computation of $\mathcal{D}$ on $E(g_i)$ depends on the node orders. Following the common practice in graph generation [70], we characterize the decision process of $\mathcal{D}$ with a fixed graph ordering, *i.e.*, the BFS-ordering $\pi$ over edges $E(g_i)$. $\mathcal{D} = f \circ \Phi$ is composed by another GNN encoder $\Phi$ and scoring function $f$ over an edge sequence $E^\pi : \{e_1, e_2, ..., e_n\}$, which makes predictions on the BFS-ordered edges.

---

[3]Later in section 3.2, we will discuss the equivalence between MI$(g_i, z_i)$ and MI$((g_i, x_i), z_i)$ when node feature is structure-respecting.

Recall our previous definition on the direction of k-hop ego-graph, the center node encoder $\Psi$ receives pairs of $(g_i, x_i)$ while the neighbor node encoder $\Phi$ in discriminator $\mathcal{D}$ receives $(\tilde{g}_i, x_i)$. Both encoders are parameterized as GNNs,

$$\Psi(g_i, x_i) = \text{GNN}_\Psi(A_i, X_i), \Phi(\tilde{g}_i, x_i) = \text{GNN}_\Phi(A_i', X_i),$$

where $A_i, A_i'$ is the adjacency matrix with self-loops of $g_i$ and $\tilde{g}_i$, respectively. The self-loops are added following the common design of GNNs, which allows the convolutional node embeddings to always incorporate the influence of the center node. $A_i = A_i'^\mathsf{T}$. The output of $\Psi$, $i.e.$, $z_i \in \mathbb{R}^n$, is the center node embedding, while $\Phi$ outputs representation $H \in \mathbb{R}^{|g_i| \times n}$ for neighbor nodes in the ego-graph.

Once node representation $H$ is computed, we now describe the scoring function $f$. For each of the node pair $(p, q) \in E^\pi$, $h_p$ is the source node representation from $\Phi$, $x_q$ is the destination node features. The scoring function is,

$$f(h_p, x_q, z_i) = \sigma \left( U^T \cdot \tau \left( W^T [h_p || x_q || z_i] \right) \right), \tag{2}$$

where $\sigma$ and $\tau$ are Sigmoid and ReLU activation functions. Thus, the discriminator $\mathcal{D}$ is asked to distinguish a positive $((p, q), z_i)$ and negative pair $((p, q), z_i'))$ for each edge in $g_i$.

$$\mathcal{D}(g_i, z_i) = \sum_{(p,q) \in E^\pi} \log f(h_p, x_q, z_i), \quad \mathcal{D}(g_i, z_i') = \sum_{(p,q)}^{E^\pi} \log f(h_p, x_q, z_i'). \tag{3}$$

There are two types of edges $(p, q)$ in our consideration of node orders, *type-a* - the edges across different hops (from the center node), and *type-b* - the edges within the same hop (from the center node). The aforementioned BFS-based node ordering guarantees that Eq. 3 is sensitive to the ordering of type-a edges, and invariant to the ordering of type-b edges, which is consistent with the requirement of our theoretical analysis on $\Delta_\mathcal{D}$. Due to the fact that the output of a k-layer GNN only depends on a k-hop ego-graph for both encoders $\Psi$ and $\Phi$, EGI can be trained in parallel by sampling batches of $g_i$'s. Besides, the training objective of EGI is transferable as long as $(g_i, x_i)$ across source graph $G_a$ and $G_b$ satisfies the conditions given in §3.2. More model details in Appendix §B and source code in the Supplementary Materials.

**Connection with existing work.** To provide more insights into the EGI objective, we also present it as a dual problem of ego-graph reconstruction. Recall our definition of ego-graph mutual information $\text{MI}(g_i, \Psi(g_i, x_i))$. It can be related to an ego-graph reconstruction loss $R(g_i | \Psi(g_i, x_i))$ as

$$\max \text{MI}(g_i, \Psi(g_i, x_i)) = H(g_i) - H(g_i | \Psi(g_i, x_i)) \leq H(g_i) - R(g_i | \Psi(g_i, x_i)). \tag{4}$$

When EGI is maximizing the mutual information, it simultaneously minimizes the upper error bound of reconstructing an ego-graph $g_i$. In this view, the key difference between EGI and VGAE [28] is they assume each edge in a graph to be observed independently during the reconstruction. While in EGI, edges in an ego-graph are observed jointly during the GNN decoding. Moreover, existing mutual information based GNNs such as DGI [54] and GMI [41] explicitly measure the mutual information between node features $x$ and GNN output $\Psi$. In this way, they tend to capture node features instead of graph structures, which we deem more essential in graph transfer learning as discussed in §3.2.

**Use cases of EGI framework.** In this paper, we focus on the classical domain adaption (direct-transferring) setting [7], where no target domain labels are available and transferability is measured by the performance discrepancy without fine-tuning. In this setting, the transferability of EGI is theoretically guaranteed by Theorem 3.1. In §4.1, we validated this with the airport datasets. Beyond direct-transferring, EGI is also useful in the more generalized and practical setting of transfer learning with fine-tuning, which we introduced in §4.2 and validated with the YAGO datasets. In this setting, the transferability of EGI is not rigorously studied yet, but is empirically shown promising.

**Supportive observations.** In the first three columns of our synthetic experimental results (Table 1), in both cases of transfering GNNs between similar graphs (F-F) and dissimilar graphs (B-F), EGI significantly outperforms all competitors when using node degree one-hot encoding as transferable node features. In particular, the performance gains over the untrained GIN show the effectiveness of training and transfering, and our gains are always larger than the two state-of-the-art unsupervised GNNs. Such results clearly indicate advantageous structure preserving capability and transferability of EGI.

## 3.2 Transferability analysis based on local graph Laplacians

We now study the transferability of a GNN (in particular, with the training objective of $\mathcal{L}_{\mathrm{EGI}}$) between the source graph $G_a$ and target graph $G_b$ based on their graph similarity. We firstly establish the requirement towards node features, under which we then focus on analyzing the transferability of EGI *w.r.t.* the structural information of $G_a$ and $G_b$.

Recall our view of the GNN output as a combination of its input node features, fixed graph Laplacian and learnable graph filters. The utility of a GNN is determined by the compatibility among the three. In order to fulfill such compatibility, we require the node features to be *structure-respecting*:

**Definition 3.3** (Structure-respecting node features)**.** *Let $g_i$ be an ordered ego-graph centered on node $v_i$ with a set of node features $\{x^i_{p,q}\}^{k,|V_p(g_i)|}_{p=0,q=1}$, where $V_p(g_i)$ is the set of nodes in p-th hop of $g_i$. Then we say the node features on $g_i$ are structure-respecting if $x^i_{p,q} = [f(g_i)]_{p,q} \in \mathbb{R}^d$ for any node $v_q \in V_p(g_i)$, where $f : \mathcal{G} \to \mathbb{R}^{d \times |V(g_i)|}$ is a function. In the strict case, $f$ should be injective.*

In its essence, Def 3.3 requires the node features to be a function of the graph structures, which is sensitive to changes in the graph structures, and in an ideal case, injective to the graph structures (*i.e.*, mapping different graphs to different features). In this way, when the learned graph filters of a transfered GNN is compatible to the structure of $G$, they are also compatible to the node features of $G$. As we will explain in Remark 2 of Theorem 3.1, this requirement is also essential for the analysis of EGI transferability which eventually only depends on the structural difference between two graphs.

In practice, commonly used node features like node degrees, PageRank scores [40], spectral embeddings [11], and many pre-computed unsupervised network embeddings [42, 51, 14] are all structure-respecting in nature. However, other commonly used node features like random vectors [68] or uniform vectors [60] are not and thus non-transferable. When raw node attributes are available, they are transferable as long as the concept of *homophily* [36] applies, which also implies Def 3.3, but we do not have a rigorous analysis on it yet.

**Supportive observations.** In the fifth and sixth columns in Table 1, where we use same fixed vectors as non-transferable node features to contrast with the first three columns, there is almost no transferability (see $\delta(acc.)$) for all compared methods when non-transferable features are used, as the performance of trained GNNs are similar to or worse than their untrained baselines. More detailed experiments on different transferable and non-transferable features can be found in Appendix §C.1.

With our view of graphs and requirement on node features both established, now we derive the following theorem by characterizing the performance difference of EGI on two graphs based on Eq. 1.

**Theorem 3.1** (GNN transferability)**.** *Let $G_a = \{(g_i, x_i)\}^n_{i=1}$ and $G_b = \{(g_{i'}, x_{i'})\}^m_{i'=1}$ be two graphs, and assume node features are structure-relevant. Consider GCN $\Psi_\theta$ with k layers and a 1-hop polynomial filter $\phi$. With reasonable assumptions on the local spectrum of $G_a$ and $G_b$, the empirical performance difference of $\Psi_\theta$ evaluated on $\mathcal{L}_{\mathrm{EGI}}$ satisfies*

$$|\mathcal{L}_{\mathrm{EGI}}(G_a) - \mathcal{L}_{\mathrm{EGI}}(G_b)| \leq \mathcal{O}\left(\Delta_{\mathcal{D}}(G_a, G_b) + C\right). \tag{5}$$

*On the RHS, $C$ is only dependent on the graph encoders and node features, while $\Delta_{\mathcal{D}}(G_a, G_b)$ measures the structural difference between the source and target graphs as follows,*

$$\Delta_{\mathcal{D}}(G_a, G_b) = \tilde{C} \frac{1}{nm} \sum_{i=1}^{n} \sum_{i'=1}^{m} \lambda_{\max}(\tilde{L}_{g_i} - \tilde{L}_{g_{i'}}) \tag{6}$$

*where $\lambda_{\max}(A) := \lambda_{\max}(A^T A)^{1/2}$, and $\tilde{L}_{g_i}$ denotes the normalised graph Laplacian of $\tilde{g}_i$ by its in-degree. $\tilde{C}$ is a constant dependant on $\lambda_{\max}(\tilde{L}_{g_i})$ and $\mathcal{D}$.*

*Proof.* The full proof is detailed in Appendix §A. $\qquad\square$

The analysis in Theorem 3.1 naturally instantiates our insight about the correspondence between structural similarity and GNN transferability. It allows us to tell how well an EGI trained on $G_a$ can work on $G_b$ by only checking the local graph Laplacians of $G_a$ and $G_b$ without actually training any model. In particular, we define the *EGI gap* as $\Delta_{\mathcal{D}}$ in Eq. 6, as other term $C$ is the same for different methods using same GNN encoder. It can be computed to bound the transferability of EGI regarding its loss difference on the source and target graphs.

**Remark 1.** *Our view of a graph $G$ as samples of k-hop ego-graphs is important, as it allows us to obtain node-wise characterization of GNN similarly as in [55]. It also allows us to set the depth of ego-graphs in the analysis to be the same as the number of GNN layers (k), since the GNN embedding of each node mostly depends on its k-hop ego-graph instead of the whole graph.*

**Remark 2.** *For Eq. 1, Def 3.3 ensures the sampling of GNN embedding at a node always corresponds to sampling an ego-graph from $\mathcal{G}$, which reduces to uniformly sampling from $G = \{g_i\}_{i=1}^n$ under the setting of Theorem 3.1. Therefore, the requirement of Def 3.3 in the context of Theorem 3.1 guarantees the analysis to be only depending on the structural information of the graph.*

**Supportive observations.** In Table 1, in the $\bar{d}$ columns, we compute the average structural difference between two Forest-fire graphs ($\Delta_\mathcal{D}$(F,F)) and between Barabasi and Forest-fire graphs ($\Delta_\mathcal{D}$(B,F)), based on the RHS of Eq. 5. The results validate the topological difference between graphs generated by different random-graph models, while also verifying our view of graph as k-hop ego-graph samples and the way we propose based on it to characterize structural information of graphs. We further highlight in the $\delta$(acc) columns the accuracy difference between the GNNs transfered from Forest-fire graphs and Barabasi graphs to Forest-fire graphs. Since Forest-fire graphs are more similar to Forest-fire graphs than Barabasi graphs (as verified in the $\Delta_\mathcal{D}$ columns), we expect $\delta$(acc.) to be positive and large, indicating more positive transfer between the more similar graphs. Indeed, the behaviors of EGI align well with the expectation, which indicates its well-understood transferability and the utility of our theoretical analysis.

**Use cases of Theorem 3.1.** Our Theorem 3.1 naturally allows for two practical use cases among many others: *point-wise pre-judge* and *pair-wise pre-selection* for EGI pre-training. Suppose we have a target graph $G_b$ which does not have sufficient training labels. In the first setting, we have a single source graph $G_a$ which might be useful for pre-training a GNN to be used on $G_b$. The EGI gap $\Delta_\mathcal{D}(G_a, G_b)$ in Eq. 6 can then be computed between $G_a$ and $G_b$ to pre-judge whether such transfer learning would be successful before any actual GNN training (*i.e.*, yes if $\Delta_\mathcal{D}(G_a, G_b)$ is empirically much smaller than 1.0; no otherwise). In the second setting, we have two or more source graphs $\{G_a^1, G_a^2, \ldots\}$ which might be useful for pre-training the GNN. The EGI gap can then be computed between every pair of $G_a^i$ and $G_b$ to pre-select the best source graph (*i.e.*, select the one with the least EGI gap).

In practice, the computation of eigenvalues on the small ego-graphs can be rather efficient [2], and we do not need to enumerate all pairs of ego-graphs on two compared graphs especially if the graphs are really large (*e.g.*, with more than a thousand nodes). Instead, we can randomly sample pairs of ego-graphs from the two graphs, update the average difference on-the-fly, and stop when it converges. Suppose we need to sample $M$ pairs of k-hop ego-graphs to compare two large graphs, and the average size of ego-graphs are $L$, then the overall complexity of computing Eq. 5 is $\mathcal{O}(ML^2)$, where $M$ is often less than 1K and $L$ less than 50. In Appendix §C.4, we report the approximated $\Delta_\mathcal{D}$'s *w.r.t.* different sampling frequencies, and they are indeed pretty close to the actual value even with smaller sample frequencies, showing the feasible efficiency of computing $\Delta_\mathcal{D}$ through sampling.

**Limitations.** EGI is designed to account for the structural difference captured by GNNs (*i.e.*, k-hop ego-graphs). The effectiveness of EGI could be limited if the tasks on target graphs depend on different structural signals. For example, as Eq. 6 is computing the average pairwise distances between the graph Laplacians of local ego-graphs, $\Delta_\mathcal{D}$ is possibly less effective in explicitly capturing global graph properties such as numbers of connected components (CCs). In some specific tasks (such as counting CCs or community detection) where such properties become the key factors, $\Delta_\mathcal{D}$ may fail to predict the transferability of GNNs.

## 4  Real Data Experiments

**Baselines.** We compare the proposed model against existing self-supervised GNNs and pre-training GNN algorithms. To exclude the impact of different GNN encoders $\Psi$ on transferability, we always use the same encoder architecture for all compared methods (*i.e.*, GIN [60] for direct-transfering experiments, GCN [29] for transfering with fine-tuning).

The self-supervised GNN baselines are GVAE [28], DGI [54] and two latest mutual information estimation methods GMI [41] and MVC [17]. As for pre-training GNN algorithms, MaskGNN

**Table 1:** Synthetic experiments of identifying structural equivalent nodes. We randomly generate 40 graphs with the Forest-fire model (F) [32] and 40 graphs with the Barabasi model (B) [1], The GNN model is GIN [60] with random parameters (baseline with only the neighborhood aggregation function), VGAE[28], DGI [54], and EGI with GIN encoder. We train VGAE, DGI and EGI on one graph from either set (F and B), and test them on the rest of Forest-fire graphs (F). Transferable feature is node degree one-hot encoding and non-transferable feature is uniform vectors. More details about the results and dataset can be found in Appendix §C.1

| Method | transferable features | | | non-transferable feature | | | structural difference | |
|---|---|---|---|---|---|---|---|---|
| | F-F | B-F | $\delta$(acc.) | F-F | B-F | $\delta$(acc.) | $\Delta_{\mathcal{D}}$(F,F) | $\Delta_{\mathcal{D}}$(B,F) |
| GIN (untrained) | 0.572 | 0.572 | / | 0.358 | 0.358 | / | | |
| VGAE (GIN) | 0.498 | 0.432 | +0.066 | 0.240 | 0.239 | 0.001 | | |
| DGI (GIN) | 0.578 | 0.591 | -0.013 | 0.394 | 0.213 | +0.181 | 0.752 | 0.883 |
| EGI (GIN) | **0.710** | 0.616 | +0.094 | 0.376 | 0.346 | +0.03 | | |

and ContextPredGNN are two node-level pre-training models proposed in [21] Besides, Structural Pre-train [23] also conducts unsupervised node-level pre-training with structural features like node degrees and clustering coefficients.

**Experimental Settings.** The main hyperparameter $k$ is set 2 in EGI as a common practice. We use Adam [27] as optimizer and learning rate is 0.01. We provide the experimental result with varying $k$ in the Appendix §C.4. All baselines are set with the default parameters. Our experiments were run on an AWS g4dn.2xlarge machine with 1 Nvidia T4 GPU. By default, we use node degree one-hot encoding as the transferable feature across all different graphs. As stated before, other transferable features like spectral and other pre-computed node embeddings are also applicable. We focus on the setting where the downstream tasks on target graphs are unspecified but assumed to be structure-relevant, and thus pre-train the GNNs on source graphs in an unsupervised fashion.[4] In terms of evaluation, we design two realistic experimental settings: (1) Direct-transfering on the more structure-relevant task of role identification without given node features to directly evaluate the utility and transferability of EGI. (2) Few-shot learning on relation prediction with task-specific node features to evaluate the generalization ability of EGI.

## 4.1 Direct-transfering on role identification

First, we use the role identification without node features in a *direct-transfering* setting as a reliable proxy to evaluate transfer learning performance regarding different pre-training objectives. Role in a network is defined as nodes with similar structural behaviors, such as *clique members*, *hub* and *bridge* [19]. Across graphs in the same domain, we assume the definition of role to be consistent, and the task of role identification is highly structure-relevant, which can directly reflect the transferability of different methods and allows us to conduct the analysis according to Theorem 3.1. Upon convergence of pre-training each model on the source graphs, we directly apply them to the target graphs and further train a multi-layer perceptron (MLP) upon their outputs. The GNN parameters are frozen during the MLP training. We refer to this strategy as *direct-transfering* since there is no fine-tuning of the models after transfering to the target graphs.

We use two real-world network datasets with role-based node labels: (1) Airport [45] contains three networks from different regions– Brazil, USA and Europe. Each node is an airport and each link is the flight between airports. The airports are assigned with external labels based on their *level of popularity*. (2) Gene [68] contains the gene interactions regarding 50 different cancers. Each gene has a binary label indicating whether it is a *transcription factor*. More details about the results and dataset can be found in Appendix C.2.

The experimental setup on the Airport dataset closely resembles that of our synthetic experiments in Table 1, but with real data and more detailed comparisons. We train all models (except for the untrained ones) on the Europe network, and test them on all three networks. The results are presented in Table 2. We notice that the node degree features themselves (with MLP) show reasonable performance in all three networks, which is not surprising since the popularity-based airport role labels are highly relevant to node degrees. The untrained GIN encoder yields a significant margin over just node features, as GNN encoder incorporates structural information to node representations.

---

[4]The downstream tasks are unspecified because we aim to study the general transferability of GNNs that is not bounded to specific tasks. Nevertheless, we assume the tasks to be relevant to graph structures.

While training of the DGI can further improve the performance on the source graph, EGI shows the best performance there with the structure-relevant node degree features, corroborating the claimed effectiveness of EGI in capturing the essential graph information (*i.e.* recover the k-hop ego-graph distributions) as we stress in §3.

When transfering the models to USA and Brazil networks, EGI further achieves the best performance compared with all baselines when structure relevant features are used (64.55 and 73.15), which reflects the most significant positive transfer. Interestingly, direct application of GVAE, DGI and MVC that do not capture the input k-hop graph jointly, leads to rather limited and even negative transferrability (through comparison against the untrained GIN encoders). The recently proposed transfer learning frameworks for GNN like MaskGNN and Structural Pre-train are able to mitigate negative transfer to some extent, but their performances are still inferior to EGI. We believe this is because their models are prone to learn the graph-specific information that is less transferable across different graphs. GMI is also known to capture the graph structure and node features, so it achieves second best result comparing with EGI.

Similarly as in Table 1, we also compute the structural differences among three networks *w.r.t.* the EGI gap in Eq. 6. The structural difference is 0.869 between the Europe and USA networks, and 0.851 between the Europe and Brazil datasets, which are pretty close. Consequently, the transferability of EGI regarding its performance gain over the untrained GIN baseline is $4.8\%$ on the USA network and $4.4\%$ on the Brazil network, which are also close. Such observations again align well with our conclusion in Theorem 3.1 that the transferability of EGI is closely related to the structural differences between source and target graphs.

**Table 2:** Results of role identification with direct-transfering on the Airport dataset. We report mean and standard deviation over 100 runs. The scores marked with ** passed t-test with $p < 0.01$ over the second runners.

| Method | Airport [45] | | |
|---|---|---|---|
| | Europe | USA | Brazil |
| features | 0.528±0.052 | 0.557±0.028 | 0.671±0.089 |
| GIN (random-init) | 0.558±0.050 | 0.616±0.030 | 0.700±0.082 |
| GVAE (GIN) [28] | 0.539±0.053 | 0.555±0.029 | 0.663±0.089 |
| DGI (GIN) [54] | 0.578±0.050 | 0.549±0.028 | 0.673±0.084 |
| Mask-GIN [21] | 0.564±0.053 | 0.608±0.027 | 0.667±0.073 |
| ContextPred-GIN [21] | 0.527±0.048 | 0.504±0.030 | 0.621±0.078 |
| Structural Pre-train [23] | 0.560±0.050 | 0.622±0.030 | 0.688±0.082 |
| MVC [17] | 0.532±0.050 | 0.597±0.030 | 0.661±0.093 |
| GMI [41] | 0.581±0.054 | 0.593±0.031 | 0.731±0.107 |
| EGI (GIN) | **0.592±0.046**** | **0.646±0.029** ** | **0.732±0.078** |

On the Gene dataset, with more graphs available, we focus on EGI to further validate the utility of Eq. 5 in Theorem 3.1, regarding the connection between the EGI gap (Eq. 6) and the performance gap (micro-F1) of EGI on them. Due to severe label imbalance that removes the performance gaps, we only use the seven brain cancer networks that have a more consistent balance of labels. As shown in Figure 3, we train EGI on one graph and test it on the other graphs. The $x$-axis shows the EGI gap, and $y$-axis shows the improvement on micro-F1 compared with an untrained GIN. The negative correlation between two quantities is obvious. Specifically, when the structural difference is smaller than 1, positive transfer is observed (upper left area) as the performance of transferred EGI is better than untrained GIN, and when the structural difference becomes large ($> 1$), negative transfer is observed. We also notice a similar graph pattern, *i.e.* single dense cluster, between source graph and positive transferred target graph $G_2$.

## 4.2 Few-shot learning on relation prediction

Here we evaluate EGI in the more generalized and practical setting of *few-shot learning* on the less structure-relevant task of relation prediction, with task-specific node features and fine-tuning. The source graph contains a cleaned full dump of 579K entities from YAGO [49], and we investigate 20-shot relation prediction on a target graph with 24 relation types, which is a sub-graph of 115K entities sampled from the same dump. In *post-fine-tuning*, the models are pre-trained with an unsupervised loss on the source graph and fine-tuned with the task-specific loss on the target graph. In *joint-fine-tuning*, the same pre-trained models are jointly optimized *w.r.t.* the unsupervised pre-training loss

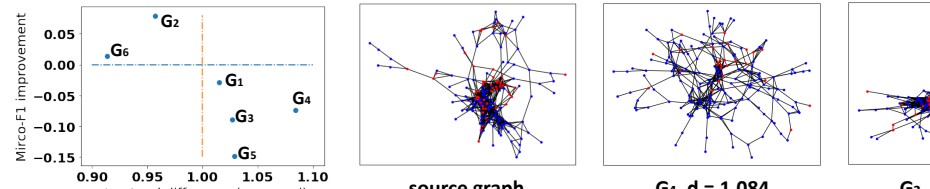

**Figure 3:** Transfer learning performance of role identification on the Gene dataset. We visualize the source graph $G_0$ and two example target graphs that are relatively more different ($G_4$) or similar ($G_2$) with $G_0$.

and task-specific fine-tuning loss on the target graph. In Table 3, we observe most of the existing models fail to transfer across pre-training and fine-tuning tasks, especially in the *joint-fine-tuning* setting. In particular, both Mask-GIN and ContextPred-GIN rely a lot on task-specific fine-tuning, while EGI focuses on the capturing of similar ego-graph structures that are transferable across graphs. The mutual information based method GMI also demonstrates considerable transferability and we believe the ability to capture the graph structure is the key to the transferability. As a consequence, EGI significantly outperforms all compared methods in both settings. More detailed statistics and running time are in Appendix §C.3.

**Table 3:** Performance of few-shot relation prediction on YAGO. The scores marked with $**$ passed t-test with $p < 0.01$ over the second best results.

| Method | post-fine-tuning | | joint-fine-tuning | |
|---|---|---|---|---|
| | AUROC | MRR | AUROC | MRR |
| No pre-train | 0.687±0.002 | 0.596±0.003 | N.A. | N.A. |
| GVAE | 0.701±0.003 | 0.601±0.007 | 0.679±0.004 | 0.568±0.008 |
| DGI | 0.689±0.011 | 0.586±0.025 | 0.688±0.012 | 0.537±0.023 |
| MaskGNN | 0.713±0.009 | 0.631±0.015 | 0.712±0.005 | 0.560±0.010 |
| ContextPredGNN | 0.692±0.030 | 0.662±0.030 | 0.705±0.011 | 0.575±0.021 |
| GMI | 0.728±0.005 | 0.625±0.009 | 0.721±0.007 | 0.643±0.011 |
| Structural Pre-train | OOM | OOM | OOM | OOM |
| MVC | OOM | OOM | OOM | OOM |
| EGI | **0.739± 0.009**$^{**}$ | **0.670±0.014** | **0.787 ± 0.011**$^{**}$ | **0.729 ± 0.016**$^{**}$ |

## 5   Conclusion

To the best of our knowledge, this is the first research effort towards establishing a theoretically grounded framework to analyze GNN transferability, which we also demonstrate to be practically useful for guiding the design and conduct of transfer learning with GNNs. For future work, it is intriguing to further strengthen the bound with relaxed assumptions, rigorously extend it to the more complicated and less restricted settings regarding node features and downstream tasks, as well as analyze and improve the proposed framework over more transfer learning scenarios and datasets. It is also important to protect the privacy of pre-training data to avoid potential negative societal impacts.

## Acknowledgments and Disclosure of Funding

Research was supported in part by US DARPA KAIROS Program No. FA8750-19-2-1004, SocialSim Program No. W911NF-17-C-0099, and INCAS Program No. HR001121C0165, National Science Foundation IIS-19-56151, IIS-17-41317, and IIS 17-04532, and the Molecule Maker Lab Institute: An AI Research Institutes program supported by NSF under Award No. 2019897. Chao Zhang is supported NSF IIS-2008334, IIS-2106961, and ONR MURI N00014-17-1-2656. We would like to thank AWS Machine Learning Research Awards program for providing computational resources for the experiments in this paper. This work is also partially supported by the internal funding and GPU servers provided by the Computer Science Department of Emory University. Any opinions, findings, and conclusions or recommendations expressed herein are those of the authors and do not necessarily represent the views, either expressed or implied, of DARPA or the U.S. Government.

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
