# OpenReview forum: "Transfer Learning of Graph Neural Networks with Ego-graph Information Maximization"
_NeurIPS.cc/2021/Conference — NeurIPS 2021 Poster_

### Official Review · Reviewer_2Ymd · 2021-07-10

**Rating:** 6
**Confidence:** 5

**Summary:**

This paper develops a novel measure for assessing the transferability of
graph neural network models to new data sets. The measure is based on
a decomposition of graphs into 'ego networks' (essentially,
a distribution of $k$-hop subgraph, extracted from a given larger
graph). Transferability is then assessed by means of a spectral
criterion using the graph Laplacian. Experiments demonstrate the utility
in assessing transferability in such a manner, as the new measure
appears to be aligned with improvements in predictive performance.

**Limitations And Societal Impact:**

Algorithmic limitations are partially discussed in this paper, but as
outlined above, an ablation study of the effects of changing parameters
and/or sampling frequency is crucial and needs to be included. In
addition, 'failure cases' of the algorithm would also be interesting to
discuss; are there specific graphs that turn out be easily transferable
but the framework is unable to assess this a priori?

There are no specific adverse societal impacts arising from this work.
The paper briefly mentions privacy aspects in pre-training data, but
this thought could be fleshed out more.

**Main Review:**

## Summary of the Review

This paper discusses a timely and highly relevant topic. Understanding
*a priori* how a graph neural network will be able to perform on unseen
data is an excellent research objective; addressing this problem is
bound to lead to better models, and might even make 'pre-trained model
markets' more attractive for GNNs.

While I overall feel favourably about the paper, there are several
issues that need to be addressed before endorsing it for publication.
These comprise:

1. Unclear limitations or properties of the developed measure:

  - The main text is currently not discussing how choices of $k$
    affect the calculation. If I understood everything correctly, $k
    = 2$ is set for all experiments; a more thorough analysis of
    different values of $k$ is necessary. It is clear that higher values
    of $k$ will not be beneficial with respect to computational
    performance, but their impact still needs to be assessed. I would be
    interested in learning to what extent transferability becomes more
    'precise' or more 'predictive' by these changes.

  - To what extent is the $\Delta_{\mathcal{D}}$ dependent on
    the ordering of input nodes? The way I understand Equation 6 is that
    two normalised graph Laplacians (which are of the same size because
    the same value for $k$ is being used throughout the paper) are
    subtracted from each other. However, this presupposes that the
    ordering of nodes is known in advance, which I do not think is the
    case here. This aspect needs to be discussed. (there is a brief
    discussion on edge ordering, following Eq. 1; if this applies here
    as well, I would suggest to mention it explicitly)

  - How is EGI used in practice? Do I understand this correctly that
    it is being employed as an additional loss term? Figure 2 seems to
    suggest at first that the primary purpose is to assess how well
    a model could be transferred, but the experiments later on seem to
    indicate that EGI is used *during* training. Please clarify this!
    (an alternative use case, discussed in l. 237-- could involve
    picking the best graphs as a training data set; this would be a more
    'static' use of the measure, so please comment on the exact details
    here)

2. Missing clarity, for instance when it comes to defining what the
   method is supposed to capture. The paper uses the term 'graph
   structures,' but seems to equate this with $k$-hop graphs. Upon
   re-reading the paper, I find that a more appropriate description
   would be that similarity of graph Laplacians is supposed to be
   captured (with the caveat about the dependence on a fixed ordering
   that I listed above).

3. The experimental setup could be improved: please report mean accuracy
   and standard deviations over multiple runs. This would help in
   assessing the proposed method.

## Comments on clarity

- Why does Definition 3.1 make use of a *directed* distance formulation
  between graph nodes? The remainder of the paper appears to discuss
  only the undirected case.

- Why is the second input to $\mathrm{sp}(\cdot)$ negated in Equation 1?
  The function $\mathcal{D}$ returns a probability, right?

- Why are self-loops required or included here? (l. 143)

- Please provide more details about the arguments $h_p$, $x_q$, and
  $z_i$ shown in Eq. 2. I find the expression hard to read at present.

- Definition 3.3 appears to state that structure-respecting node
  features are those features that are function of the $k$-hop graphs.
  Is this correct? If so, please provide this intuition in the
  definition; I found it confusing at first.

- What does it mean to be 'injective to the graph structures'? I can
  guess at the intent, namely that $f$ should map different $k$-hop
  graphs to different representations/features. Is this correct?

- I would not use the term 'organic node attributes' but rather discuss
  that certain graphs might already be equipped with node attributes.

- In l. 226--, a brief depiction of such graphs in the main text would
  be useful to make the paper more accessible to readers.

- Please also discuss the stability of your measure and method with
  respect to random samples. I found the section starting in l. 247
  very interesting; the statements made in it should also be empirically
  evaluated.

- When displaying results, such as in Table 1, consider sorting columns
  differently. I would find it more intuitive to *first* see the
  'original' accuracy obtained on the 'source' data set, whereas the
  *second* column could depict the 'target' data set.

- For Table 2, is 'EGI' also using a GIN architecture or a GCN? Please
  mention the architecture directly in the 'Method' column.

- Please provide more details on data set features in Table 2.

## Minor Comments

- I disagree with the in the related work that 'GNNs are not very
  complicated.' Maybe the convolutional layer can be seen as
  a comparatively simple construction, but I would rephrase or tone down
  this section slightly.

- For the discussion of graph kernels, the author may want to consider
  citing one of the recent surveys in this area, such as 'Graph Kernels:
  State-of-the-Art and Future Challenges ' by Borgwardt et al., 'A
  Survey on Graph Kernels' by Kriege et al., or 'Graph Kernels:
  A Survey' by Nikolentzos et al.

## Style & language

The language in this paper requires some improvements. I had trouble
understanding certain constructions, and I would suggest to rewrite them
in order to improve the accessibility of the paper. Here are some
suggestions:

- As an overall suggestion, I would like to point out that bibliographic
  citations can be sorted using the `sort` option of the `natbib`
  package, for instance. I find that this makes parsing citations
  a little bit easier.

- 'on the' appears twice in the caption of Figure 1

- 'its capability' --> 'their capability'

- 'none of existing' --> 'none of the existing'

- 'training objective is' --> 'training objectives are'

- 'set of $k$-hop ego graph' --> 'set of $k$-hop ego graphs'

- 'drawn with the joint distribution' --> 'drawn from the joint distribution'

- 'from empirical joint' --> 'drawn from an empirical joint'

- 'we now describes' --> 'we now describe'

- 'distinguish positive' --> 'distinguish a positive'

- 'make node-wise characterization' --> 'obtain node-wise characterizations'

- 'while also verify' --> 'while also verifying'

- 'hyper parameter' --> 'hyperparameter'

- 'apply them on' --> 'apply them to'

- 'parameters are freezing' --> 'parameters are frozen'

- 'are prune' --> 'are prone'

- 'structural different' --> 'structural differences'

- 'that vanishes the performance gaps' --> 'that removes [...]'

# Update after rebuttal

The authors provided numerous clarifications, which helped alleviate some of my initial concerns. I am therefore happy to raise my score accordingly.

**Time Spent Reviewing:**

5

---

> ### Author Response · Authors · 2021-08-10
> **Initial response to Reviewer 2Ymd**
>
> We thank the reviewer for valuable and detailed feedback, which acknowledges our accomplishments and provides many opportunities for further improvements. Following are our detailed responses and revision plans.
>
> ## 1. Additional experimental analysis and results.
> ### 1.1 On the choices of $k$ (towards the precision and predictiveness of transferability).
> In principle, $k$ may affect the transferability of EGI in two ways: (1) larger $k$ may make the EGI model more expressive (better precision) and the EGI gap more accurate (better predictiveness); (2) However, the GNN encoders may suffer from the over-smoothing problem and the $\Delta_D$ computations may suffer from more noises. Therefore, it is hard to determine the influence of $k$ without empirical analysis. We thank the reviewer for pointing this out and have conducted additional experiments on our airport datasets. The results are as follows. As we can observe, when $k=1$ or $k=3$, the classification accuracy of the source graph is worse than $k=2$, likely because the GNN encoder is either less powerful or over-smoothed. As a result, $k=2$ obtains the best transferability to both the USA and Brazil networks. When $k=3$, $\Delta_D$ likely accounts for too subtle/noisy ego-graph differences and may become less effective in predicting the transferability.
>
> |                      |Europe (source) |    USA (target)    |          Brazil (target)|
> | ------------------| --------------------| -------------------------| ------------------------------|
> |      Method |acc             |    acc, $\Delta_D$  |      acc, $\Delta_D$|
> |EGI (k=1)        |    58.25         |    60.08, 0.385         |  60.74, 0.335|
> |EGI (k=2)      |      59.15      |       64.55, 0.869       |     73.15, 0.851  |
> |EGI (k=3)       |     57.63       |     64.12, 0.912       |    72.22, 0.909|
>
>
>
>
> ### 1.2 On the sampling frequency (towards the computation of $\Delta_D$)
>
> For the stability of the EGI gap when computed through samping, we have added experiments of $\Delta_D$ vs. sampling frequency (10-round) on our airport datasets. More rigorous study on its convergence is also an interesting future direction. As we can observe, large sample frequency leads to more accurate and robust estimation of $\Delta_D$.
>
> |Sampling frequency | Europe-USA | Europe-Brazil |
> |-----------|---------------|---------------|
> |100 pairs  |                  0.872±0.039 |     0.854±0.042|
> |1000 pairs    |              0.859±0.012   |   0.848±0.007|
> |Full             |                0.869       |          0.851|
>
>
> ### 1.3 Standard deviations of the results.
>
> For Table 2, the results with std (100-round) inserted are as follows. The large variance is due to the small number of testing nodes, which is consistent with existing studies on the same datasets [1].
>
> | Method       |  Europe     | USA     | Brazil   |
> |---------------|----------------|-----------|------------|
> |Features  | 0.528±0.052  |  0.557±0.028  | 0.671±0.089 |
> | GIN(untrained)  |  0.558±0.050 |  0.616±0.030 |   0.700±0.082 |
> | GVAE  |          0.539±0.053  | 0.555±0.029  | 0.663±0.089 |
> | DGI   |      0.578±0.050 |  0.549±0.028 |  0.673±0.084 |
> | MaskGNN  | 0.564±0.053  | 0.608±0.027 |  0.667±0.073 |
> | ContextPredGNN  | 0.527±0.048 |   0.504±0.030  | 0.621±0.078 |
> | Structural Pre-train |  0.560±0.050  |  0.622±0.030 |  0.688±0.082 |
> | MVC  | 0.532±0.050  | 0.597±0.030  | 0.661±0.093 |
> | GMI  | 0.581±0.054  | 0.593±0.031 |  0.731±0.107 |
> | EGI  | 0.592±0.046  | 0.646±0.029  | 0.732±0.078 |
>
>
>
> For Table 3, due to large runtimes, we have only finished the 10-round experiments on EGI and the second runner GMI so far. We will continue the experiments and add the complete results in a revision.
>
> *Setting: post-fine-tuning*
>
> |    Method                | AUROC  |  MRR   |
> |  ----------------| ------------|-----------|
> GMI |0.7279±0.0052 | 0.6472±0.0093
> EGI |0.7389±0.0087 | 0.6695±0.0139
>
> *Setting: joint-fine-tuning*
>
> |    Method                | AUROC  |  MRR   |
> |  ----------------| ------------|-----------|
> GMI |0.7122±0.0069 | 0.6270±0.0113
> EGI |0.7870±0.0107 | 0.7289±0.0164
>
>
>
> ## 2. Clarity
> ### 2.1 EGI use cases.
>
> As recognized by the reviewer, our work mainly includes two contributions, the EGI objective and the EGI theory. For the EGI objective, we briefly talked about the direct-transferring setting in l.98-99, which is very close to the classical domain adaptation setting with non-graph data [2]. In this setting, the transferability of EGI is theoretically guaranteed, and in Section 4.1, we validated this with the airport datasets. Beyond direct-transferring, the EGI objective is also useful in the more generalized and practical setting of transfer learning with fine-tuning, which we introduced in Section 4.2 and validated with the YAGO datasets. In this setting, the transferability of EGI is not rigorously studied yet, but is empirically shown promising. As also suggested by Reviewer 2, we think it is good to have a small paragraph between l.154 and l.155 to explicitly summarize these use cases. As for the EGI theory, we explained its intended use cases in the paragraph of l.237, which is based on the proved bound in Theorem 3.1, and validated through the observations of the actual positive correlations between $\Delta_D$ and performance gaps in both synthetic data (Table 1) and real data (mainly Figure 2).
>
> ### 2.2 Dependence on node orders.
>
> There are two types of node ordering in our consideration of ego-graphs, (1) the node orders across different hops (from the center node), and (2) the node orders within the same hop (from the center node). In Eq.(6), $\Delta_D$ is sensitive to (1) and invariant to (2), due to the proposition [3, P.76] that eigenvalues of a product of square matrices are invariant under cyclic permutation of the product order. As a consequence, in the EGI objective, we use the BFS ordering because we care about (1) but not (2). We made a typo on edge ordering under Eq.(1) where we actually meant node ordering, which we will definitely fix in a revision.
>
> ### 2.3 EGI failure cases.
>
> As Eq.(6) is computing the average pairwise distances between the graph Laplacians of local ego-graphs, $\Delta_D$ is possibly less effective in explicitly capturing global graph properties such as numbers of connected components (CCs). In some specific tasks (such as counting CCs) where such properties become the key factors, our $\Delta_D$ may fail to predict the transferability of GNNs, and our EGI objective may be ineffective in capturing such properties.
>
> ### 2.4 Other smaller issues.
>
> 2.4.1 We defined the structural information of graphs in Definition 3.2 based on ego-graph distributions, but we agree that it is more appropriate to exactly explain that the “graph structure” we aim to capture with $\Delta_D$ and EGI is the similarity of graph Laplacians with a BFS node ordering.
>
> 2.4.2 We used directed ego-graphs in Definition 3.1 because we want to clearly define the BFS node ordering that is important both for the computation of $\Delta_D$ and the EGI objective. However, we realized that this only needs to be specified on directed graphs, and applies trivially to undirected graphs with $g=\tilde{g}$, which are the actual cases in our experiments. We will clarify this in a revision.
>
> 2.4.3 The second input to $sp(\cdot)$ in Eq.(1) is negated because $(g_i, z_i)$ denotes a positive node-neighborhood pair, while $(g_i, z’_i)$ denotes a negative pair.
>
> 2.4.4 The inclusion of self-loops in l.143 is following the common design of GNNs, which allows the convolutional node embeddings to always incorporate the influence of the center node.
>
> 2.4.5 We will provide exact equations for $h$, $z$ and $x$ around Eq.(2) in a revision.
>
> 2.4.6 We will provide the intuition of structure-respecting node features in Definition 3.3 in a revision.
>
> 2.4.7 It is very correct that “injective to the graph structures” means mapping different k-hop graphs to different features, and we will clarify this in a revision.
>
> 2.4.8 When using “organic node attributes”, we were exactly referring to node attributes that are already in the graphs. We may have misunderstood the meaning of “organic”, and will revise it accordingly.
>
> 2.4.9 We have some visualizations of such synthesized graphs in the appendix, and we will include some brief intuitive descriptions of them in l.226 in a revision.
>
> 2.4.10 The results shown in Table 1 are all on target datasets (with models trained on different source datasets). We will clarify this in a revision.
>
> 2.4.11 We will specify the GIN encoder of EGI in Tables 2 and 3 as we did for Table 1.
>
> 2.4.12 The features we used for the datasets in Table 2 are node degree one-hot encodings and uniform vectors of all 1’s. We will clarify this in a revision.
>
> ## 3. Other issues
>
> We sincerely thank the reviewer for providing such detailed suggestions for the improvements of our paper, and will take the advice to do the following in a revision.
> * We will tone down the discussion about the complication of GNNs.
> * We will add the references to related works on graph kernels.
> * We have carefully checked every piece of suggestions in the style and language section, and will apply them as fit in a revision.
>
> ## References
> [1] Ribeiro, Leonardo FR, Pedro HP Saverese, and Daniel R. Figueiredo. "struc2vec: Learning node representations from structural identity." KDD. 2017.
> [2] Shai Ben-David, John Blitzer, Koby Crammer, and Fernando Pereira. “Analysis of representations for domain adaptation.” NIPS. 2007.
> [3] Anton, Howard, and Chris Rorres. “Elementary linear algebra: applications version.” John Wiley & Sons, 2013.

---

> > ### Comment · Reviewer_2Ymd · 2021-08-16
> > **Thanks**
> >
> > Thanks for clarifying my concerns! Trusting the authors that they will incorporate these changes, I will raise my score accordingly. One additional question from my side: how are the standard deviations calculated? Do they use the same units as the accuracy in the tables? I am asking because sdev seems to be surprisingly small.

---

> > > ### Author Response · Authors · 2021-08-16
> > > **Thanks again**
> > >
> > > We thank the reviewer again for accepting our responses and trusting us in further improving this work. For the reviewer's additional question, the standard deviations are calculated after 10-round experiments, using the same units as the accuracy in the tables. The std's are in fact not that small according to our understanding.
> > >
> > > Particularly, the ones on the airport datasets are usually around 0.05, and reaches as high as 0.09 and 0.1 in some cases, given the fact that the accuracy differences among algorithms are usually at the scale of 0.01. This is consistent with previous studies such as [1] (e.g., Figure 7) in our additional references above. The ones on the YAGO datasets are smaller, since the testing datasets are much larger.

---

### Official Review · Reviewer_5DSQ · 2021-07-13

**Rating:** 7
**Confidence:** 3

**Summary:**

This paper proposes a novel graph transfer learning framework with theoretical guarantees. Theoretical analysis is supported by both synthetic and real-world experiments.

**Main Review:**

Originality: The work is moderately original with ego-graph information maximization scheme.

Quality: The transferability analysis seems theoretically sound and solid with improved transferability bounds. Experiments are carried out based on real-world datasets. I do not see any major issues with this work.

Clarity: The paper is well-organized overall. For clarity, I have some small concerns: In line 226, there is no reference to $\bar d$ columns.

**Time Spent Reviewing:**

4

---

> ### Author Response · Authors · 2021-08-10
> **Initial response to Reviewer 5DSQ**
>
> We thank the reviewer for the comments and time spent reviewing our paper. We are glad that the reviewer appreciated our contributions in this novel framework of GNN transferability with both theoretical analysis and empirical support. For the reviewer's question on clarity, in l.227 we explained $\bar{d}$ as the average $\Delta_D$ between one source and multiple target synthetic graphs to ensure fairness in the evaluation.

---

### Official Review · Reviewer_7fvg · 2021-07-16

**Rating:** 6
**Confidence:** 2

**Summary:**

The paper establishes a theoretically grounded and practically useful framework for transfer learning of GNNs.

**Main Review:**

The work seems complete, with both theoretical guidance and promising experimental results. The main drawback is that it is densely written (unclear in some parts) and difficult to follow. The authors are strongly encouraged to improve the readability of the paper, especially the methodology part. I have a few comments and questions.

 - Why does the computation of $\mathcal{D}$ on $E(g_i)$ depend on the edge orders (line 139)? According to Eq.3, the summation over edges does not seem to depend on edge orders.
 - What is the exact definition of the discriminator $\mathcal{D}$? It seems that both $f$ and $\Phi$ are a part of $\mathcal{D}$, but how are they combined together to form $\mathcal{D}$?
 - The topic is about transfer learning, but I cannot figure out how the model transfers from the source graph $G_a$ to the target graph $G_b$. It should be presented between Line 154 and 155.
 - Calling $\Phi$ decoder is confusing. The input of $\Phi$ is not the output of the encoder $\Psi$, so they are not a pair of encoder and decoder, but rather both are encoders.


**Time Spent Reviewing:**

7 hours

---

> ### Author Response · Authors · 2021-08-10
> **Initial response to Reviewer 7fvg**
>
> We thank the reviewer for the time and effort in generating the valuable feedback, which acknowledges our complete contributions on theoretical studies and empirical analysis, and provides suggestions on the improvements of technical writing. Following are our clarifications and plans for improvements.
>
> ## 1. The exact definition of the discriminator $\mathcal{D}$.
> In this work, we propose EGI as an unsupervised GNN training objective towards effective and guaranteed transferability. The main difference between EGI and existing unsupervised GNN training objectives such as GVAE[1] and DGI [2] is in our novel design of the discriminator $\mathcal{D}$, where we focus on the mutual information between the nodes and their neighborhoods (defined by the ego-graphs). Therefore, in its essence, $\mathcal{D}$ is maximizing a score between the node and its actual neighborhood (positive neighborhood), while minimizing that score between the node and any other node’s neighborhood (negative neighborhood).
>
> ### 1.1 What is $\Phi$ and should it be called a decoder?
> To enable the computation of the score between a node and its (positive/negative) neighborhood, we need to first learn the representations of the node and the neighborhood. For the node, we directly use the representation computed by $\Psi$, whereas for the neighborhood, we use the representations of all nodes in it computed by $\Phi$. We realized that it is indeed more appropriate to also call $\Phi$ an encoder and we thank the reviewer for the suggestion. The reason to compute two encoders $\Psi$ and $\Phi$ is mostly due to model performance, which is motivated by the design of target/context embeddings in popular frameworks such as the Skipgram [3]. To further differentiate $\Psi$ and $\Phi$, we will call them the center node encoder and neighbor node encoder, respectively and further clarify this design in a revision.
>
> ### 1.2 How are $f$ and $\Phi$ combined in $\mathcal{D}$?
> The reviewer’s understanding is correct that our discriminator $\mathcal{D}$ involves both the scoring function $f$ and the neighbor node encoder $\Phi$, as well as the center node encoder $\Psi$. As shown in Eq.(3), the scoring function $f$ is applied on the triplets of (1) center node embedding $z$ (computed by $\Psi$), (2) neighbor node embedding $h$ (computed by $\Phi$), and (3) original node features $x$.
>
> ### 1.3 Why and how is edge ordering captured in $\mathcal{D}$?
> We made a typo on edge ordering under Eq.(1) where we actually meant node ordering, which we will definitely fix in a revision. There are two types of node ordering in our consideration of ego-graphs, (1) the node orders across different hops (from the center node), and (2) the node orders within the same hop (from the center node). In EGI, we use the BFS ordering because we care about (1) but not (2), which is due to the intended consistency with our theoretical studies based on local graph Laplacians, where the computation of $\Delta_D$ is dependent on (1) but independent on (2) (more details in Appendix A). In Eq.(3) in the main paper, this is achieved because we are summarizing over all node pairs $(p, q)$ in the BFS tree $E^{\pi}$, which only includes edges going from the $k$th hop to the $k+1$th (across hops). This Eq.(3) is exactly sensitive to (1), as the interchange of node order on any pair of $(p, q)$ will influence the value of that pair, and in the meantime invariant to (2), as the summation ignores the orders of particular pairs of $(p, q)$’s. We thank the reviewer for pointing this out and will also add clarifications about this in a revision.
>
> ## 2. The use cases of EGI.
> We briefly talked about the direct-transferring setting in l.98-99, which is very close to the classical domain adaptation setting with non-graph data [4]. In this setting, the transferability of EGI is theoretically guaranteed, and in Section 4.1, we validated this with the airport datasets. Beyond direct-transferring, EGI is also useful in the more generalized and practical setting of transfer learning with fine-tuning, which we introduced in Section 4.2 and validated with the YAGO datasets. In this setting, the transferability of EGI is not rigorously studied yet, but is empirically shown promising. We explained the use cases of the EGI theory in the paragraph of l.237, but did not explicitly summarize the use cases of the EGI loss. We agree it is better to have a small paragraph between l.154 and l.155 exactly for this purpose, and we thank the reviewer for this great suggestion.
>
> ## References
> [1] Kipf, Thomas N., and Max Welling. "Variational graph auto-encoders." NIPS: Bayesian Deep Learning Workshop (2016).
> [2] Veličković, Petar, et al. "Deep graph infomax." ICLR (2019).
> [3] Mikolov, Tomas, et al. "Distributed representations of words and phrases and their compositionality." NIPS (2013).
> [4] Shai Ben-David, John Blitzer, Koby Crammer, and Fernando Pereira. “Analysis of representations for domain adaptation.” NIPS (2007).

---

> > ### Comment · Reviewer_7fvg · 2021-08-24
> > **Please improve the clarity of writing**
> >
> > Thank you for the response. I am willing to upgrade my initial rating to 6 (but with lower confidence). Please do improve the clarity of writing in the final version.

---

> > > ### Author Response · Authors · 2021-08-25
> > > **Thanks. We will definitely improve the technical writing.**
> > >
> > > Dear Reviewer 7fvg,
> > >
> > > We are grateful for your reconsideration. Thanks to your questions, we are more aware of the proper improvements to be made on the technical clarity, and will do our best to achieve them in the revision.

---

### Official Review · Reviewer_RQmo · 2021-07-16

**Rating:** 8
**Confidence:** 4

**Summary:**

This work aims to provide fundamental understanding towards the mechanism and transferability of GNNs, and develops an unsupervised GNN training objective based on their understanding. Novel theoretical analysis has been done to support the design of EGI and establish its transferability bound, while the effectiveness of EGI and the utility of the transferability bound are verified by extensive experiments. The whole story looks new, comprehensive and convincing to me.

**Limitations And Societal Impact:**

See main review

**Main Review:**

Most arguments made and techniques developed in this work are well-supported by solid and detailed theoretical analysis. However, I do find a few limitations of this work. For example, (1) its focus on “structure-respecting features” might lead to the exclusion of its usage in many real-world networks where node features are more noisy and even not “structure-respecting”; (2) its study of transferability under the “direct-transfer” setting might be less useful when in real-world, most transfer learning is done with fine-tuning. Nonetheless, I understand the necessarily existing gap between theory and practice in many machine learning problems, and I appreciate the authors in making their assumptions clear and also conducting empirical analysis to show the generalizability of their model and theory over the scenarios where the assumptions are less valid.

The paper is overall well-organized, and I like the intermediate “supportive observations” in the technical section which help the understanding and verification of the technical contributions. In terms of writing, the introduction is very clear, while some details in the technical section require some back-and-forth reading. Further polishing over that section would benefit the overall readability.

To the best of my knowledge, this paper is the first one that provides a well-balanced blend of theory and empirical studies aiming to demystify the transfer learning with GNNs. Although its current scope is a bit limited, I think it is a necessary and good start for future research in this important direction.


**Time Spent Reviewing:**

10

---

> ### Author Response · Authors · 2021-08-10
> **Initial responses to Reviewer RQmo**
>
> We appreciate the positive feedback on our theoretical and empirical analysis towards the transferability of GNNs. The reviewer also brings up two potential future directions: (1) theoretical studies on the transferability bound with relaxed assumptions on node features and downstream tasks, and (2) empirical studies on the more practical settings of GNN transferring such as with fine-tuning. We agree these are both very important follow-up directions.
>
> Currently, we are carefully working on the clarification of possible ambiguities in the draft, and actively applying the EGI framework in emerging real-world applications with limited training data, such as brain network modeling and healthcare network predictions.

---

### Decision · Program_Chairs · 2021-09-27

**Decision:**

Accept (Poster)

**Comment:**

This paper studies, in the context of transfer learning, the extent to which trained graph neural networks can transfer to new data. Concretely, the authors derive criteria to asses the transferability of a based on ego-graph information (EGI) maximization. In initial reviews, most reviewers marked the paper above acceptance threshold, and following rebuttal and discussion we have reached a unanimous decision to recommend its acceptance.

Some remarks have been raised (especially by one reviewer) regarding the clarity and quality of presentation, but in discussion a consensus has been reached that the revisions required to address these concerned can viably be regarded as only minor revisions that should not prevent the paper from being accepted, especially given the author responses that indicated they will revise the manuscript appropriately.

Therefore, I recommend this paper be accepted and reiterate to the authors to carefully go over the reviewer comments and follow up on the indicated or planned revisions from their responses.